# Ingestion of Carbohydrate Prior to and during Maximal, Sprint Interval Cycling Has No Ergogenic Effect: A Randomized, Double-Blind, Placebo Controlled, Crossover Study

**DOI:** 10.3390/nu12082223

**Published:** 2020-07-25

**Authors:** Gerard McMahon, Aaron Thornbury

**Affiliations:** Sport & Exercise Sciences Research Institute, School of Sport, Ulster University, Newtownabbey, Belfast BT37 0QB, UK; Thornbury-A1@ulster.ac.uk

**Keywords:** anaerobic, ergogenic aid, solution, substrate utilisation

## Abstract

Carbohydrate (CHO) ingestion may improve intermittent sprint performance in repeated sprint efforts ≤15 s. Yet, evidence for its efficacy on sprint interval durations ~30 s is lacking. The purpose of this study was to investigate the effects of CHO ingestion on maximal sprint interval exercise. Fifteen (*n* = 15) recreational athletes (13/2 males/females, age 22 ± 2 years; height 176 ± 11 cm; mass 76.8 ± 11.3 kg) volunteered for this randomised, double-blind, placebo-controlled, crossover design. Participants completed two experimental trials (performed 10-days apart) involving the ingestion of an 8% CHO solution or a flavour and appearance-matched placebo (PLA) solution (5 mL/kg/bw), immediately before exercise, and preceding the second interval of four × 30 s bouts of repeated maximal sprint efforts (separated by 3.5 min of passive recovery). Peak and mean power (W) output progressively decreased during the repeated sprints (main effect of time, *p* < 0.0001), but there were no differences between CHO and PLA during any of the sprints (*p* > 0.05 for condition main effect and condition × time interaction). Physiological responses (blood lactate, heart rate, oxygen consumption, respiratory exchange ratio and RPE) were also unaltered by CHO ingestion. In conclusion, CHO ingestion does not enhance performance or modulate physiological responses during intermittent maximal, sprint cycling.

## 1. Introduction

It is well documented that acute carbohydrate (CHO) supplementation can enhance exercise performance across a range of exercise intensities and durations [1,2]. For example, CHO ingestion can help preserve blood glucose concentrations and augment rates of glucose oxidation in the later stages of prolonged exercise, when glycogen stores in skeletal muscle and liver are depleted [3]. It has also been shown that performance during shorter, higher intensity continuous exercise [2], and during high-intensity intermittent running [4,5,6] can be enhanced with carbohydrate supplementation compared to a placebo. Yet, less is known about CHO supplementation during maximal sprint interval exercise (SIE).

SIE is the performance of exercise at an ‘all-out’ or ‘supramaximal’ effort, separated by periods of active or passive recovery [7], and has been shown to be a potent stimulus to promote whole-body and skeletal muscle oxidative adaptations in both recreationally active and highly trained populations [8,9,10,11]. There is evidence to suggest that both exercise intensity and exercise volume are the two primary components in modulating increases in aerobic adaptations, such as mitochondrial content, through high intensity interval training [7,12]. Therefore, and firstly from a training perspective, there is the potential that the addition of exogenous CHO prior to or during sprint interval exercise may: allow an individual to exercise to exercise more intensely (rate of work done), or similarly, complete more work within a specified timeframe (i.e., greater volume of work completed). If this greater exercise intensity/volume is repeated over a sustained period of training, this could potentially lead to superior physiological adaptations, such as skeletal muscle oxidative capacity, ${\dot{V}O}_{2}$ max and endurance performance capacity [12,13,14]. Secondly, from a performance perspective, there are several sports/events, such as high performance cycling, where maximising the amount of work done (i.e., mean power output) can lead to improved performance outcomes [15,16]. Eight out of the 28 Union Cycliste Internationale (UCI) championship races are all-out sprint events, with four events decided upon by final sprint efforts, and two events requiring repeated sprints. As such, sprint ability is a major performance determinant for many events [17].

To date, there are only a few studies that have investigated the effects of CHO ingestion (prior to and during exercise) on repeated maximal SIE performance and associated physiological responses. These previous studies have reported a beneficial effect of acute CHO ingestion on repeated sprint performance versus placebo [18], versus CHO and caffeine combined [19], and versus CHO mouth rinse [20] conditions. Previous recommendations have been made to suggest CHO ingestion may have an ergogenic effect on exercise durations lasting a minimum 45–60 min [21], and that ergogenic effects of CHO of exercise durations lasting 30–75 min can be mediated via the nervous system using CHO mouth-rinse strategies [2,21]. More recently, it has been demonstrated that the ingestion of 120 g/h of CHO can attenuate internal exercise load and also improve post-exercise recovery, by reducing exercise-induced muscle damage following a trail marathon [22,23]. However, such recommendations have been made in the context of continuous, time-trial or intermittent team sport type performance tests, and have not considered repeated, maximal efforts of a shorter total duration. Each of the three aforementioned SIE studies reporting an ergogenic effect of CHO ingestion employed a variety of different maximal SIE protocols, including 5 × 15 s maximal sprints interspersed with 4 min of active recovery [20], 6 × 5 s maximal sprints interspersed with 25 secs passive recovery [18], and 10 sets of 5 × 4 s sprints with 20 s active recovery between sprints, and 2 min between sets [19]. Therefore, the total session durations of these protocols were 17.25 min, 2.5 min and 34.66 min respectively, with extremely low actual exercise durations of 75 s, 30 s and 200 s, respectively. These studies therefore highlight the potential of CHO ingestion as an ergogenic aid in exercise durations of considerably less than 30 min.

Moreover, there are a multitude of different protocol designs used during SIE, which are likely reflective of the intent to induce different (acute) physiological stresses, through altering the contribution of the ATP-PC, glycolytic and aerobic energy systems (and therefore accrual of different metabolites and enzymes involved from said systems [24,25,26]), ultimately, for either adaptive or performance replication outcomes. However, possibly the most frequently used SIE protocol for anaerobic/aerobic training involves several consecutive 30 s maximal efforts, separated by ≥ 3 min of passive recovery [8,10,11,14]. In terms of performance, 30 s all out efforts reflect specific cycling sprint durations in the Team Sprint (M2 rider), Women’s 500 m time trials, and the final sprint in the Keirin. To date, no studies have investigated the role of CHO supplementation with interval durations of 30 s. The purpose of the present investigation was to assess the effects of acute CHO supplementation during repeated, longer duration, maximal sprint interval exercise, on indices of cycling performance and associated physiological responses.

## 2. Materials and Methods

### 2.1. Participants

Fifteen physically active males (*n* = 13) and females (*n* = 2) (mean±SD: age: 22 ± 2 yrs; height: 176 ± 11 cm; mass: 76.8 ± 11.3 kg, BMI 24.7 ± 2.4 kg/m^2^) volunteered to participate in the study, and were recruited from the local university campus using posters, e-mails and word of mouth. To be eligible for the study, individuals must have been aged between 18–39 years old, and had to participate in moderate intensity physical activity at least 2–3 times per week. All participants took part in recreational athletic endeavours, e.g., university or amateur club level sports. Self-reported activity levels of participants were 2–3 training sessions of moderate-high intensity activity lasting ~90 min each per week, in addition to 1–2 competitive matches per week. This varied slightly depending on each individual participant’s sport and point in their relative competitive season. Exclusion criteria included not having any musculoskeletal or neurological disorders, free from injury and not currently supplementing with any ergogenic aids either 3 months prior to or during the study. Following a pre-screening physical activity questionnaire to ensure eligibility, participants were provided with information sheet, outlining the full experimental procedure and risks involved. All participants gave their written informed consent to participate. The study was conducted in accordance with the Declaration of Helsinki, and the protocol was approved by the Ulster University School of Sport Ethics Committee (MSc321 2018–2019).

### 2.2. Study Design

The study utilised a double-blind, placebo-controlled, randomised, cross-over design, with two experimental conditions: carbohydrate supplementation and placebo. Participants each visited the laboratory on two occasions to perform a bout of SIE (4 × 30 s maximal sprints). All participants self-reported as being familiar with SIE through routine training in their sports. Ten minutes prior to and during the bout of SIE, participants consumed either a carbohydrate solution or a placebo (described below). The order of conditions was randomised with ~10 days washout between the two trials. All participants were instructed to refrain from exercise 48 h prior to their lab visit and to abstain from alcohol and caffeine for 24 h before testing. All participants verbally confirmed on the morning of the trial that they avoided exercise for 48 h and consumption of alcohol and caffeine for 24 h. Participants were asked to maintain an identical nutritional intake 24 h before each visit to the lab, with participants’ last food intake (breakfast) ~3 h prior to exercise. Participants confirmed that they had maintained nutritional intakes in the 24-h period prior to exercise between trials at their second laboratory visit. The timing of each trial was standardised within and between participants, and the lab conditions were similar (temperature between 19–21 °C) during each visit. A member of the research team prepared each of the solutions and provided them to the participants, reaffirming the ingestion instructions. Neither the researcher recording/analysing performance and physiological variables, nor the participant, knew the contents of the solution. The blinding continued to the stage of statistical analysis, where the trial conditions were revealed.

### 2.3. Experimental Protocol

Participants reported to the laboratory where physical characteristics were assessed. Height and body mass were measured using a stadiometer with integrated scales (SECA, Birmingham, UK). Body mass was recorded on each of the two laboratory visits. Participants then performed a 5 min warm-up on a cycle ergometer (Wattbike Pro, Wattbike, Nottingham, UK) at a self-selected intensity, perceived to be 10 on the Borg scale (i.e., ‘light/fairly light’). Pre-exercise blood lactate and heart rate were taken at rest ~3–5 min, following completion of the warm-up and approximately 1 min prior to SIE. Participants then performed four bouts of SIE, consisting of repeated maximal effort for 30 s at a standardised air resistance of level 6, followed by 3.5 min of passive recovery following each interval. During recovery, participants were allowed to disembark the ergometer, but remain stationary beside it, with no walking etc. permitted. Participants ingested a carbohydrate solution consisting of 8% carbohydrate (100% Maltodextrin, MyProtein, THG, Manchester, UK) and a single lemon-flavoured sweetener tablet per litre water (Splenda, Heartland Food Products Group, PA, USA), or a placebo consisting of water and a single lemon sweetener tablet per litre. The volume of solution ingested was 5 mls/kg/body mass. Exactly 50% of the solution was consumed approximately ten minutes prior to the first interval, with the remaining solution to be ingested during the rest periods of the first and second interval. To reduce the possibility of participants identifying the beverage they were consuming, both condition solutions were lemon-flavoured, colourless and presented in opaque bottles. An exit question was posed upon completion of the study to ascertain whether participants detected any differences in solutions between trials. All participants reported being unable to identify any differences between the experimental solutions. Performance and physiological measures were recorded during each sprint interval. Verbal encouragement was provided during each experimental trial, to encourage the participants to maintain maximal effort on the cycle ergometer. Each session lasted approximately 30 min in total.

### 2.4. Performance Outcomes

The primary outcomes of interest in this study were peak and mean power output during each 30 s sprint. Power output was measured (100 Hz) continuously during each sprint (Wattbike Pro, Wattbike, Nottingham, UK), saved, and subsequently downloaded for analysis using the WattBike Expert Software. Relative peak power output (RPPO) and relative mean power output (RMPO) were calculated by dividing the absolute power outputs by body mass.

### 2.5. Physiological Outcomes

Physiological responses, including oxygen uptake, respiratory exchange ratio (RER), blood lactate concentrations, heart rate, and ratings of perceived exertion (RPE) were secondary outcomes of this study.

### 2.6. Oxygen Uptake and Respiratory Exchange Ratio

To measure oxygen uptake (${\dot{V}O}_{2}$) and RER, participants were fitted with a silicone face mask connected to a metabolic cart, with an online gas analysis system to enable breath by breath analysis of expired air (Cosmed, Quark CPET, Rome, Italy). The metabolic cart was calibrated prior to each participant trial. Breath by breath measurements of ${\dot{V}O}_{2}$ and RER were recorded from 60 s before the first 30 s sprint and terminated upon cessation of the final 30 s sprint. Only the data from each of the 30 s exercise intervals were analysed.

### 2.7. Blood Lactate Quantification

To obtain blood lactate, an alcohol wipe (Alcotip swab, Universal Hospital Supplies, London, UK) was used to cleanse the participant’s finger, and a sterile lancet (Accu Chek, Roche, Mannheim, Germany) was used to pierce the skin of the finger to produce blood. A Lactate Pro 2 lactate strip (Arkray, KDK Corp., Shiga, Japan) collected the blood, which was then inserted into a Lactate Pro 2^TM^ lactate analyser (Arkray, KDK Corp., Shiga, Japan), to determine the lactate content of the participant’s blood. Lactate measures were taken during the first 30 s, following the completion of the interval.

### 2.8. Heart Rate and RPE

Heart rate was recorded using a Polar H7 heart rate monitor (Polar, Kempele, Finland) and a Polar RS400 watch, and as taken immediately, following the cessation of the interval. RPE (rate of perceived exertion) was provided by each participant via the Borg scale, and was taken immediately following cessation of the interval.

### 2.9. Statistical Analysis

All data were analysed using SPSS software version 25 (IBM, Armonk, NY, USA). A Shapiro–Wilk test was used to check each variable for any deviation from a normal distribution. Data were revealed to be normally distributed and parametric. A two-way (condition × time) repeated measures ANOVA, with sex (male/female) included as a covariate, was applied to determine the effect of CHO supplementation compared to placebo on performance and physiological outcomes during SIE. If appropriate, post-hoc comparisons were analysed via paired sample *t*-tests with a Bonferroni correction. In order to assess potential order effects on performance variables, a two-way (condition × time) repeated measures ANOVA was applied, with trial order as the between factor [27]. Statistical significance was set a priori at *p* < 0.05. The effect size for the main effects and interactions was estimated by calculating partial eta squared values (η_p_^2^), with 0.01, 0.06 and 0.14 used to denote small, moderate and large effects sizes, respectively. All data are reported as mean ± SD.

## 3. Results

Participant numbers for all results are *n* = 15, except for ${\dot{V}O}_{2}$ and RER variables (*n* = 13). There were no significant condition × time interactions assessing for order effects (*p* > 0.05) in either of the performance variables. A paired *t*-test revealed body mass was not different between trials (Trial 1; 77.0 ± 11.1 Kg, Trial 2; 76.7 ± 11.3 Kg (*p* > 0.05).

### 3.1. Performance

For RPPO, during the sprints, there was a main effect of time (*p* < 0.0001, η_p_^2^ = 0.85), but no effect of condition (*p* = 0.79, η_p_^2^ = 0.03), and no condition × time interaction (*p* = 0.84, η_p_^2^ = 0.35). RPPO progressively decreased with each repeated sprint, and this effect did not appear to be impacted by CHO supplementation (Figure 1). Similarly, in terms of RMPO, there was also a main effect of time (*p* < 0.0001, η_p_^2^ = 0.83), but no effect of condition (*p* = 0.79, η_p_^2^ = 0.03) or condition × time interaction (*p*= 0.77, η_p_^2^ = 0.41). There were marked reductions in RMPO with each interval, however, this effect was not altered by CHO supplementation (Figure 2).

### 3.2. Physiological Responses

#### 3.2.1. Oxygen Uptake and Respiratory Exchange Ratio

There was a significant main effect of time on ${\dot{V}O}_{2}$ (*p* < 0.0001, η_p_^2^ = 0.88). Despite significant (*p* < 0.05) increases in ${\dot{V}O}_{2}$ during INT1 to INT 4 compared to pre-exercise (Table 1), there was no effect of condition (*p* = 0.78, η_p_^2^ = 0.01) or condition × time interaction (*p* = 0.63, η_p_^2^ = 0.11). There was a significant main effect of time on RER (*p* < 0.0001, η_p_^2^ = 0.81), but there was no effect of condition (*p* = 0.27, η_p_^2^ = 0.05) or condition × time interaction (*p* = 0.98, η_p_^2^ = 0.02). RER was increased (*p* < 0.0001) from pre-exercise compared to INT 2 and 3, with a further increase (*p* < 0.05), during INT3 compared to INT1. RER then returned to baseline levels, with a decrease (*p* < 0.05) during INT4 compared to INT3 (Table 1).

#### 3.2.2. Blood Lactate

There was a significant main effect of time for blood lactate (*p* < 0.0001, η_p_^2^ = 0.96), yet there was no evidence of effect of condition (*p* = 0.33, η_p_^2^ = 0.03) or condition × time interaction (*p* = 0.52, η_p_^2^ = 0.13). Blood lactate was significantly increased following INT1 to INT4, compared to pre-exercise (*p* < 0.0001), and systematically increased (*p* < 0.05) with each interval compared to the previous interval up until INT4 (Figure 3).

## 4. Discussion

The purpose of this placebo-controlled study was to determine whether ingestion of a carbohydrate solution immediately prior to and during exercise would affect performance or alter whole body physiological responses during SIE. The current data suggest there is no effect of CHO ingestion on any performance or physiological responses. These findings question the potential ergogenic value of CHO for repeated maximal sprint performance for SIE of longer durations than previously observed in literature. This study also provides novel insights with specific reference to performance during intervals of longer duration, which until now, have not been reported. Thus, the results of the study are important to consider when fuelling for training and performance.

### 4.1. Potential Impact of CHO on Performance

In the current study, both peak (PPO) and mean power outputs (MPO) systematically declined with each interval, relative to the previous interval, which is in agreement with observations from previous SIE studies [18,20,24,25,28]. Bogdanis et al. [25] demonstrated that following performance of a second consecutive 30 s sprint with either 1.5, 3 or 6 min of passive recovery, neither of these recovery periods were sufficient to maintain PPO or MPO compared to the first sprint. Based off this evidence, it is not entirely surprising that following the current protocol of 4 × 30 s sprints with 3.5 mins of passive recovery was insufficient to mitigate performance declines. However, CHO ingestion also did not appear to have any further effect on protecting against exercise-induced reductions in performance, compared to placebo. In order for exogenous CHO to mediate its effect as an ergogenic aid, it needs to be consumed, absorbed, delivered, taken up and preferentially oxidised over other substrates available by the skeletal muscle for use in ATP turnover. Hulston et al. [29] showed that, with a moderate (6%) CHO solution, plasma glucose and exogenous glucose oxidation were increased by 15–30 min at the onset of exercise. In the current study, 50% of an 8% solution was ingested 10 min prior to the first interval, with the remaining solution consumed before the third interval commenced. A total of 16.5 min elapsed from the beginning of sprint one until the completion of sprint four. Therefore, within the total 26.5 min timeframe, CHO may have been available for oxidation, although without mechanistic measures, this cannot be confirmed. It is also worth noting that exercise at very high intensities are a limiting factor on gastric emptying rates following glucose ingestion [30]. Therefore, it is also unclear how much of the CHO ingested would be available to the muscle for oxidation during SIE in the current study. Previous studies investigating exogenous CHO ingestion on SIE using either a 7% CHO solution consumed 30 min pre SIE [18] or 10% CHO solution consumed 5 min pre- and before each sprint [20], reported beneficial effects on performance compared to placebo or CHO mouth rinse respectively (see the next section for a further discussion). Therefore, it is entirely plausible that, if CHO ingestion mediated a performance effect in these studies, our ingestion schedule would also have provided at least the opportunity for performance enhancement. Although with recent evidence pointing to high ingestion rates of 120 g/h of CHO improving exercise performance [22,23], each of the SIE-based protocols may have benefited from a greater CHO intake. It has been shown that the phosphocreatine (PCr) and glycolytic systems are primarily responsible for the metabolism of substrates to meet the maximal demand for ATP during the first bout of maximal sprint exercise [24,25,31]. As sprint exercise progresses, muscle pH is rapidly decreased, halting glycolysis through inhibition of key glycolytic enzymes, such as phosphofructokinase (PFK), resulting in an increased demand for ATP turnover via aerobic oxidation [24,25], with increased pyruvate dehydrogenase activation and oxygen uptake, reducing lactate accumulation and increasing pyruvate oxidation [32]. Bogdanis et al. [24,25] demonstrated the mobilisation of large quantities of glycogen, with increased intramuscular glucose levels increasing substantially following initial maximal sprints. It may in fact be that preferential oxidation of existing substrates within the muscle provide sufficient energy, to fuel not only the initial bout, but also the subsequent bouts. In this case, subsequent re-oxidation of lactate and pyruvate following increased oxygen uptake during the recovery phase, is enough to fuel ATP turnover, and that the decrement in sprint performance may be more aligned to PCr recovery kinetics [24,25,28]. As such, provision of exogenous CHO in the current study, assuming delivery and availability to the muscle for oxidation, may not have been used to increase ATP turnover, and thus improve cycling performance compared to placebo.

### 4.2. Evidence for Performance Effects of CHO Ingestion and SIE

At first glance, the results of the current investigation appear to contrast with previous studies, which have concluded that CHO supplementation improves SIE performance [18,19,20]. However, there are aspects of each of these three studies which can be critiqued. For example, Lee et al. [19] found that ingestion of an 0.8 g/kg^−1^ solution had a significant performance effect on peak and mean power output, and total work versus placebo, in young, female athletes. However, on further examination, this performance effect was only evident during one of the sets (set 6), out of the total of 10 sets completed, with negligible effects observed throughout the rest of the bouts. Another study by Pomportes et al. [18] also reported a beneficial impact of acute CHO ingestion (7% CHO solution) on peak power output and perceived tiredness (a surrogate index of fatigue). However, it is worth noting that these conclusions were based on inferences made using magnitude based inferences (MBI), with ‘most likely positive’ and ‘likely positive’ effects of CHO supplementation reported for peak power and tiredness, respectively. The MBI approach has recently received substantial scrutiny for failing to adequately control type 1 error rates (i.e., false positives), particularly for effects characterised as ‘likely’ [33]. Furthermore, the study of Pomportes et al. [18] also reported (alongside MBI) no significant effect on either variable when assessed using a repeated measures ANOVA. Finally, Krings et al. [20] also investigated the influence of acute CHO ingestion (300 mL of a 10% CHO solution) on repeated maximal sprint interval performance. The results of this study reported significantly greater mean power output and total work, and significantly lower fatigue index, in the CHO ingestion condition, compared with a CHO mouth rinse condition. Critically, however, there were no differences in performance variables, compared with either an ingested or mouth rinse placebo condition, thus, inferring that other factors aside from the ingested CHO may have been associated with the performance differences between the conditions. Therefore, on the basis of these critiques, caution should be applied when interpreting such data in the context of CHO ingestion and perceived ergogenic effects on repeated sprint performance.

The current study shares some of the study design features of the three aforementioned studies that reported ergogenic effects for CHO supplementation in SIE models [18,19,20]. For example, the current study, and that of Lee et al. [19] and Krings et al. [20], were each randomised, double-blind, placebo-controlled studies, with Pomportes et al. [18] not reporting the blinding of participants to conditions. A strength of the current study’s design compared to the earlier studies was that the placebo was identical in taste, appearance, presented in opaque bottles, with all participants reporting they were unable to detect a difference between trial solutions. The studies of Pomportes et al. [18], Lee et al. [19] and Krings et al. [20] provided similar, but not identical placebo solutions to the CHO solution. Neither of these studies specifically report whether study participants were able to detect differences between solutions per condition. The description of non-identical solutions use raises the possibility that participants may have been able to identify the solution, and thus potentially confound results. The studies by Lee et al. [19], Pomportes et al. [18], and the current study reported that participants ingested breakfast meals several hours (~3 h) before the exercise protocols. However, the participants in the study of Krings et al. [20] were fasted, but no details are provided of the fasting duration. Participation in such activities in the fast stated is not reflective of ‘real-world’ practices [34], unless specifically attempting to modulate adaptations through AMPK-regulated influences on aerobic adaptations [35], although the value of SIE in these condition is questionable, in our view. Therefore, the approach used in the current study was adopted to maintain both scientific robustness [36] and real world application [34].

The novel aspect of the current study was the extended interval duration with respect to the current literature on SIE and CHO ingestion. The previous three studies employed sprint interval durations of 4, 5 and 15 s, with 20 s (passive), 25 s (passive) and 4 min (active) recovery, respectively, between sprints. The total number of sprints also ranged from 50, to 6 and 5 respectively. The current study employed 4 maximal sprints, 30 s in duration, with 3.5 min of passive recovery. The interaction of sprint interval duration with mode and duration of recovery period will result in very different energetic demands. It is likely that shorter intervals (≤15 s) derive much greater ATP turnover via PCr hydrolysis, with some contribution from glycolysis, to fuel maximal work during sprint intervals [28], whereas sprints lasting up to 30 s would derive much more ATP turnover via anaerobic glycolysis and glycogenolysis [24]. Therefore, the reliance on CHO as a fuel may have been more pertinent in the current study compared to the previous studies. Nevertheless, the volume of work completed in the study of Lee et al. [19], despite a short sprint interval duration, may also have required significant CHO contribution. Concomitantly, it is also important to consider that the total exercise durations were markedly different between the current study and previous SIE studies. In the current study, the total exercise duration was 120 s, with previous studies employing as little as 30 s [18], 75 s [20], and up to 200 s [19]. Therefore, in addition to interval length, one must also consider total volume of exercise duration when appraising the potential impact of CHO ingestion. The exogenous CHO ingested in the current study provided no additional performance benefit, and could not postpone the decline in performance during the SIE, despite other studies of shorter interval length and total exercise duration being able to report ergogenic benefits.

### 4.3. Practical Applications

The total amount of work completed (volume) or higher rate of work done (intensity), evident through the production of higher mean power outputs, may be important for adaptations associated with high-intensity interval exercise, as each of these factors have been independently shown to be associated with modulation of various adaptive signalling pathways [7,12]. Conversely, CHO ingestion may potentially attenuate cell signalling responses associated with aerobic adaptations to SIE. Guerra et al. [37] investigated the effects of either 75 g glucose ingestion (*n* = 8) versus placebo (*n* = 7), 60 min before a single maximal sprint. Interestingly, there were also no differences in relative peak or mean power between groups during the sprint, although during the recovery period, AMPKα phosphorylation was modulated in the glucose group. Despite this attenuation of AMPKα phosphorylation, there was no effect between groups on PGC-1α protein content. Therefore, the effects of glucose ingestion on such factors remain unclear. Acute ingestion of CHO did not improve mean power output in any of the intervals in the current study, nor mitigate the exercise-induced reduction in power output following the completion of further successive intervals, compared to placebo. In terms of sporting performance, many cycling events are determined by sprint performance. In the context of training and/or performance, this type of CHO ingestion does not appear to provide any ergogenic benefits.

### 4.4. Limitations

There are limitations present in the current study that warrant attention. While the participants in the current study reported being familiar with SIE through their own training and competition in sports, we did not perform a specific familiarisation with this cohort. Therefore, we cannot discard completely that there may have been some learning effects that could have added variability to the current data set. Additionally, we did not quantify/analyse/prescribe nutritional intake in the 24-h period prior to each trial, despite instructions to the participants to consume the same intake. This could have meant that glycogen stores and blood glucose levels may have been different upon commencing exercise between trials. However, as discussed already, glycogenolysis is attenuated relatively early during SIE, and performing SIE, even in a low glycogen store status, may not be a critical source of performance variability. Objective measures of participant training/fitness status were not conducted, which could affect responses and recovery to SIE, and also add variability to the data. Lastly, we did not perform any mechanistic/biochemical measurements. This meant that we were unable to detect the fate of exogenously consumed CHO and its ability to potentially alter ATP turnover in this condition. Further studies may wish to address these limitations in study design.

## 5. Conclusions

The present data suggest that acute CHO supplementation before and during repeated, 30 s maximal sprint interval exercise does not improve exercise performance or alter physiological responses compared with a placebo. We conclude that CHO ingestion immediately prior to and during short, maximal, and repeated sprint exercise is not a key consideration in a performance context or to enhance training quality.

## Figures and Tables

**Figure 1 nutrients-12-02223-f001:**
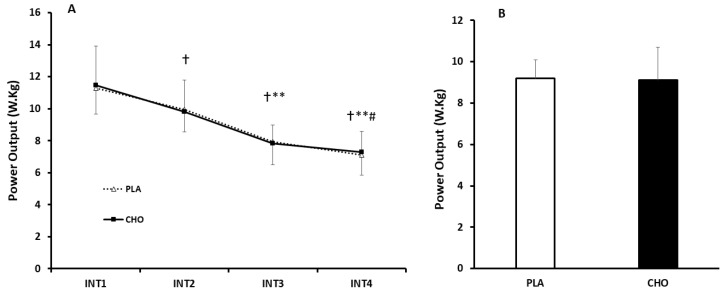
(**A**) Comparison of relative peak power output (RPPO) response between INT1–4 in acute carbohydrate (CHO) and flavour and appearance-matched placebo (PLA) trials. Data are Mean ± SD. † Significantly different to INT1 (*p* < 0.05); ** Significantly different to INT2 (*p* < 0.05) # Significantly different to INT3 (*p* < 0.05). (**B**) Aggregated RPPO across intervals. RPPO; Relative Peak Power Output.

**Figure 2 nutrients-12-02223-f002:**
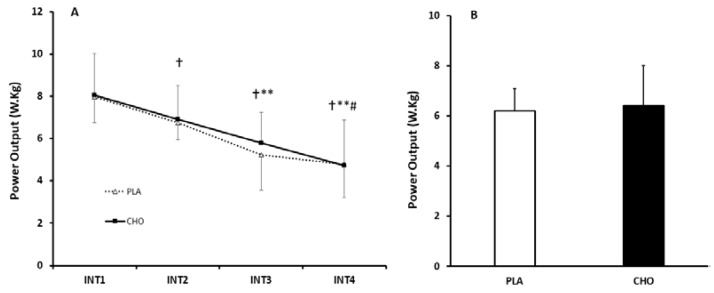
(**A**) Comparison of RMPO response between INT1–4 in CHO and PLA trials. Data are Mean ± SD. † Significantly different to INT1 (*p* < 0.05); ** Significantly different to INT2 (*p* < 0.05) # Significantly different to INT3 (*p* < 0.05). (**B**) Aggregated RMPO across intervals. RMPO; Relative Mean Power Output.

**Figure 3 nutrients-12-02223-f003:**
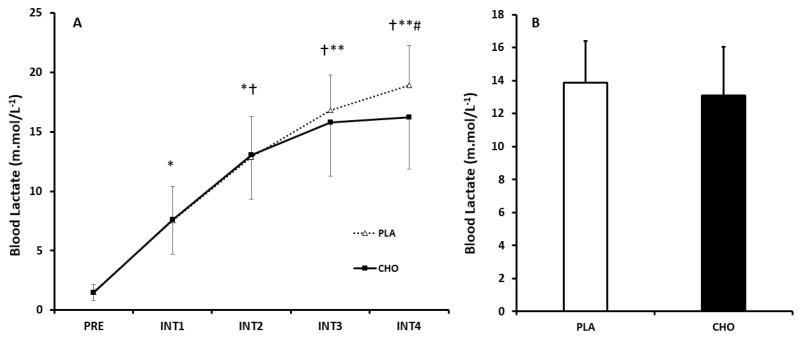
(**A**) Comparison of mean Blood Lactate response between INT1–4 in CHO and PLA trials. Data are Mean ± SD. * Significantly different to REST (*p* < 0.0001); † Significantly different to INT1 (*p* < 0.05); ** Significantly different to INT2 (*p* < 0.05) # Significantly different to INT3 (*p* < 0.05). (**B**) Aggregated blood lactate across intervals.

**Table 1 nutrients-12-02223-t001:** Physiological and Performance variables between intervals and aggregated across intervals (mean of intervals 1–4).

Variable	PRE	INT1	INT2	INT3	INT4	Aggregated
Heart Rate (bpm) PLA CHO Main Effect Time	77 ± 9 83 ± 12	164 ± 13 158 ± 13 *	171 ± 10 170 ± 7 *†	172 ± 15 171 ± 10 *†	166 ± 12 167 ± 10 *	169 ± 9 167 ± 7
${\dot{V}O}_{2}$ (mL.kg.min^−1^) PLA CHO Main Effect Time	4 ± 2 7 ± 4	32 ± 12 32 ± 15 *	29 ± 15 27 ± 14 *	27 ± 13 26 ± 12 *	31 ± 8 28 ± 10 *	29 ± 9 28 ± 11
RER PLA CHO Main Effect Time	0.88 ± 0.09 0.85 ± 0.13	0.99 ± 0.18 0.99 ± 0.18	1.09 ± 0.31 1.06 ± 0.27 *	1.20 ± 0.19 1.14 ± 0.16 *†	0.97 ± 0.25 0.95 ± 0.16 #	1.03 ± 0.07 1.00 ± 0.06
RPE (AU) PLA CHO	- -	18 ± 3 17 ± 3	18 ± 2 18 ± 2	19 ± 1 19 ± 1	20 ± 1 19 ± 2	19 ± 2 18 ± 2

Data are Mean ± SD. ${\dot{V}O}_{2}$, oxygen consumption; RER, Respiratory exchange ratio. * Significantly different to pre-exercise (*p* < 0.0001); † Significantly different to INT1 (*p* < 0.05); ** Significantly different to INT2 (*p* < 0.05); # Significantly different to INT3 (*p* < 0.05).

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
