# Peer review of "Ingestion of Carbohydrate Prior to and during Maximal, Sprint Interval Cycling Has No Ergogenic Effect: A Randomized, Double-Blind, Placebo Controlled, Crossover Study"

_nutrients, 2020, doi:10.3390/nu12082223_

Round 1

Reviewer 1 Report

Line 11-12 - suggest adding the word "repeated sprint efforts" since this is the focus of the article.

Ref 4 - I am not sure I would use a 1-hour time trial as an example of a short high intensity continuous exercise. 

For the introduction section current recommendation on carbohydrate intake during exercise for exercise of 30 min is not referred to. The reasoning for the study has not been back up appropriately by theory/evidence for exercise of 30 mins and less, duration. 

Would like to have seen carbohydrate and fluid intake for the day before exercise and for breakfast for placebo and carb-supplemented periods to be presented to ensure no differences were actually presented, even though I know participants were asked to keep the same.

The introduction section needs more evidence on why the study should realistically be completed especially considering the lack of effect of the outcome. The article has not enough evidence as to why it should be completed, and how it could actually have worked based on previous studies in the area, and current recommendations on carbohydrate intake for exercise of this duration (30 mins or less). 

In order to consider this article for publication increased reasoning as to why and how this study could actually have worked in terms of it's duration and design would be needed.

Author Response

The authors thank the reviewer for their insightful comments which have therefore improved the quality of our paper.

Line 11-12 - suggest adding the word "repeated sprint efforts" since this is the focus of the article.

The authors thank the reviewer for the suggestion and have amended accordingly

Ref 4 - I am not sure I would use a 1-hour time trial as an example of a short high intensity continuous exercise. 

The authors have reconsidered the use of this citation here and have thus removed it.

For the introduction section current recommendation on carbohydrate intake during exercise for exercise of 30 min is not referred to. The reasoning for the study has not been back up appropriately by theory/evidence for exercise of 30 mins and less, duration. 

The authors thank the reviewer for their comment. The authors have now included references to both intensity and duration of exercise to highlight the rationale for the study and appropriateness of the current study’s design.

Would like to have seen carbohydrate and fluid intake for the day before exercise and for breakfast for placebo and carb-supplemented periods to be presented to ensure no differences were actually presented, even though I know participants were asked to keep the same.

The authors thank the reviewer for their observation and whole-heartedly agree that quantifying the nutritional intake in the 24hours previous to each of the trials would have been valuable information and strengthened the design. The authors did indeed acknowledge this fact in the limitations of the original manuscript’s discussion section (L406-411 in current manuscript). The authors have also now included a line in the methods section pertaining to this issue that the participants also confirmed that they repeated their nutritional intake in the 24 hours prior to each of the trials to alleviate at least some of the concerns of the reviewer/ reader.

The introduction section needs more evidence on why the study should realistically be completed especially considering the lack of effect of the outcome. The article has not enough evidence as to why it should be completed, and how it could actually have worked based on previous studies in the area, and current recommendations on carbohydrate intake for exercise of this duration (30 mins or less). 

The authors thank the reviewer for their comment. We have now included this information in the introduction providing evidence that the three previous sprint interval exercise studies reporting beneficial effects of CHO ingestion had exercise protocol sessions lasting ~30mins or vastly less, and that actual exercise durations were extremely short including 30secs, 75secs and 200secs. The authors have also added a brief few sentences in the discussion highlighting that the exercise duration of 120secs in the current study is much greater than two of the previous studies. This highlights that the current study design was appropriate to allow an ergogenic effect of CHO ingestion to be detected if there was one, as the exercise durations are much longer than the previous studies that did report an ergogenic effect.

In order to consider this article for publication increased reasoning as to why and how this study could actually have worked in terms of its duration and design would be needed.

The authors thank the reviewer for their comments and believe that we have now provided sufficient reasoning and rationale for the current study’s duration in response to the previous questions above, where we have demonstrated that our study protocol duration (16.5mins) is considerably longer than Pomportes et al. 2016, (2.5mins) and very close in length to Krings et al. 2017 (17mins) who both report an ergogenic effect of CHO ingestion. To reiterate we have also demonstrated that the total exercise duration in the current study was 400% greater than the study of Pomportes et al. 2016 and 60% greater than Krings et al. 2017 who both reported an ergogenic effect of CHO ingestion. Taken together with our original lines of discussion in the discussion section on study design, we believe we have provided a robust argument that the current study was appropriate in terms of rationale and design for assessing the potential ergogenic effects of CHO ingestion based off evidence from previous SIE and CHO studies.

Reviewer 2 Report

Firstly, I'd like to commend the authors on a well-written paper that addresses a gap in the current literature. See comments below: 

Line 46: Suggest reword "allow an individual to exercise to produce more work done per unit of time (i.e. exercise more intensely)" for clarity. 

Line 49: Give examples of adaptations. 

Line 66: Give examples of the different acute physiological stresses. 

Line 69: Be consistent with the format of hyphens. 

Line 79: Space missing BMI 24.7±2.4 kg/m2. 

Lines 83-86: This seems unnecessarily long-winded. 

Line 143-144: The manufacturer does not need to be named twice. 

Line 151: "See below" not necessary. 

Line 187: Remove "." between S and D and be consistent throughout.

Line 193: Be consistent with spacing between figures and ±.

Lines 289-290: Change "Bogdanis and co-workers" to Bogdanis et al.

Line 293: Remove "during". 

Line: 

The limitations of the previous research in the area of CHO ingestion prior to maximal sprint interval exercise are outlined and the authors have addressed some of the major limitations i.e. an identical placebo and extended the sprint interval duration. 

Author Response

Firstly, I'd like to commend the authors on a well-written paper that addresses a gap in the current literature. See comments below: 

The authors thank the reviewer regarding their kind comments on our study

Line 46: Suggest reword "allow an individual to exercise to produce more work done per unit of time (i.e. exercise more intensely)" for clarity. 

The authors have reworded this sentence

Line 49: Give examples of adaptations. 

The authors thank the reviewer for their comment and have now provided a number of physiological adaptations associated with SIE

Line 66: Give examples of the different acute physiological stresses. 

The authors have provided the resultant stresses of different SIE protocols via outlining the reliance of different energy systems which in turn produce different metabolites and involve activity of differing proteins which then result in the unique adaptations observed with different SIE protocols.

Line 69: Be consistent with the format of hyphens. 

The authors thank the reviewer and have changed each reference to the time frames as 30-seconds to be consistent.

Line 79: Space missing BMI 24.7±2.4 kg/m2. 

Thank you, this space has been added.

Lines 83-86: This seems unnecessarily long-winded. 

The authors thank the reviewer and have broken this into two shorter sentences

Line 143-144: The manufacturer does not need to be named twice. 

Thank you this has been removed.

Line 151: "See below" not necessary.

Thank you this has been removed.  

Line 187: Remove "." between S and D and be consistent throughout.

Thank you, the authors have applied these changes throughout

Line 193: Be consistent with spacing between figures and ±.

Thank you, the authors have applied these changes throughout

Lines 289-290: Change "Bogdanis and co-workers" to Bogdanis et al.

This citation has been changed

Line 293: Remove "during". 

Thank you this has been removed.

Line: 

The limitations of the previous research in the area of CHO ingestion prior to maximal sprint interval exercise are outlined and the authors have addressed some of the major limitations i.e. an identical placebo and extended the sprint interval duration. 

Thank you, we thought it was important to highlight these limitations to add context to our findings and also in relation to other literature